# Burnout syndrome among healthcare workers during COVID-19 Pandemic in Accra, Ghana

**Kennedy Dodam Konlan** *, **Emmanuel Asampong, Phyllis Dako-Gyeke, Franklin N. Glozah**

Department of Social and Behavioural Sciences, School of Public Health, University of Ghana, Legon, Accra, Ghana

* kennedy.konlan@gmail.com

## Abstract

### Background

The emergence of the corona virus disease 2019 (COVID-19) has increased the workload of health workers particularly those in sub-Saharan Africa predisposing them to extra job-related stress and its associated job-related burnout. Burnout reduces the number, distribution and productivity of health workers. This study sought to determine personal and job-related characteristics of health workers in Accra, Ghana that influenced their experience of burnout during the COVID-19 pandemic.

### Method

A cross-sectional study was conducted among 1,264 health workers recruited from three public hospitals in Accra, Ghana between March to November, 2020. The participants were recruited using a proportionate stratified sampling technique and completed a pre-tested questionnaire that collected information on socio-demographic and job-related factors. Also, the questionnaire assessed participants' level of resilience and job-related burnout. Pearson's chi-square test was used to determine the association between burnout and the socio-demographic as well as job-related factors. However, for variables with observations less than six, a Fisher's exact test, was used to determine the associations. After the binary analysis, multivariate logistic regressions were used to determine the strength of association between the socio-demographic as well as job-related factors and burnout. Data analyses were conducted at a significant level (alpha 0.05) and power of 95% confidence with the aid of Stata 15.0.

### Results

The prevalence of burnout among the participants was 20.57% with non-clinicians displaying higher burnout compared to clinicians (26.74&% v 15.64%, p< 0.001). Health workers with 1–5 years working experience were 26.81 more likely to experience burnout (AOR = 26.81, CI = 6.37–112.9). Night shifts defined as shifts between 8:00pm to 8:00am was

**Data Availability Statement:** All relevant data are within the article and its Supporting Information files.

**Funding:** This study was supported by the Carnegie Corporation of New York (CCNY),

Building a New Generation of Academics in Africa (BANGA) Project Grant for Thesis Completion Award. The funders had no role in study design, data collection and analysis, decision to publish, or preparation of the manuscript.

associated with the 1.86 odds of experiencing burnout (OR = 1.86; 95% CI: 1.33–2.61; p<0.001). Also, participants of the primary level facility were 3.91 times more likely (AOR = 3.91, 95% CI = 2.39–6.41) to experience burnout on the job. Similarly, participants with additional jobs were 1.14 times more likely (AOR = 1.14, 95% CI = 0.75–1.74) to experience burnout. In addition, participants harboring intentions of leaving their current jobs were 4.61 times more likely (AOR = 4.61, 95% CI = 2.73–7.78) to experience burnout. Furthermore, participants with perceived high workload were 2.38 times more likely (AOR = 2.38, 95% CI = 1.40–4.05) to experience burnt-out on the job.

## Conclusion

There is high prevalence of burnout among health workers in Accra particularly during the onslaught of the COVID-19 pandemic. Working on night shifts and at the primary level of healthcare is significantly associated with increased odds of experiencing burnout.We recommend shift rotation for staff and adequate resource provision for primary level hospitals. The high burnout in this study could be influenced by other factors such as the grief caused by multiple losses and also the limited supportive resources for health workers.

## Introduction

A significant proportion of the global workforces are made up of health workers who contribute immensely towards effective healthcare delivery [1]. It is estimated that healthcare workforce represent 12% of the working population worldwide [1–3]. However, only three percent (3%) of the global health work-force are in sub-Saharan Africa (SSA) [1] and are responsible for over twenty five percent (25%) of the world's disease burden [4, 5]. Also, skilled healthcare services form the core foundation of care provisions needed to combat the global burden of maternal and child mortality [6]. With an adequately resourced healthcare system, skilled hospital staff who are trained to the requisite international standards will be capable of providing up to 90% of the maternal and new-born care needed, and can reduce maternal and new-born deaths by two-thirds [1, 6]. Furthermore, availability of health and skilled health workforce is cardinal to achieving the sustainable development goal (SDG) targets particularly those related to health [6].

Health workers consist of clinical staff (doctors, nurses/midwives, pharmacists, laboratory staff) and non-clinical staff (administrators, security staff, secretaries, managers, drivers, accountants, cooks, technicians, engineers, etc) [1, 7] who work in groups (multidisciplinary specialized team of experts) to support and assist the health and well-being of people [8]. This places a high demand on each team member thus exposing them to the risk of burnout [2]. This is further compounded in SSA where health workers are engaged in an environment that is considered as one of the most hazardous occupational settings due to the limited material and human resources for work [2, 9].

Burnout syndrome is a psycho-social disorder resulting from exposure to chronic interpersonal stress at the workplace and characterised by emotional exhaustion, negative attitude towards service recipients and a feeling of low self-accomplishment [2, 9]. Burnout only really emerged as a syndrome in the 1980s [10], when it was initially argued that primarily health workers were affected by burnout syndrome which is characterized by demanding interpersonal relationships that lead to chronic stress [5, 11]. The result of this was the depletion of

emotional and empathetic resources–emotional exhaustion–which is one of the major characteristics of burnout [12] that leaves a person feeling drained and weak. Essentially, emotional exhaustion, together with depersonalization, or cynicism, and professional incompetence, form the core features of burnout syndrome [9]. Health workers who are burnout usually have bad attitude at work [4, 10], poorly handle service recipients [3–5], and harbour intentions to quit their jobs [1, 11–14].

Recently, there have been public outcry about the attitudes of health workers in urban Ghana [8, 13] and this is evident in the recent public discourse about the phenomenon of "no bed syndrome" in health facilities [8]. Furthermore, there is evidence of poor handling and abuse of patients by healthcare staff in Accra, Ghana [15, 16] and some reasons have been advanced including job-related burnout [2, 16]. The situation has further been worsened with the emergence of COVID-19 pandemic where more people are now regular attendees of health facilities [2, 6]. The number of COVID-19 cases in Ghana rose from two (2) cases in March, 2020 to over five thousand cases (5,1667) with over three hundred (323) deaths as at the end of November 2020 [6]. Hospital staff in Accra have therefore been compelled to attend to many patients especially with the surge in positive COVID-19 cases in 2020 [2, 6] and this predisposes health workers in Accra to high workload and possible associated job-related burnout [2, 13]. This notwithstanding, there is a paucity of data on burnout syndrome among Ghanaian health workers in Accra as well as the socio-demographic and job-related factors that could be influencing the occurrence of burnout particularly during the peak of the COVID-19 pandemic in the Greater Accra Region, the region with over three-quarter of the positive cases in Ghana [6]. The rationale for this study was therefore to provide useful data on the prevalence of burnout as well as the factors influencing the experience of burnout among hospital-based staff in Accra, Ghana particularly with the onslaught of the COVID-19 pandemic [1, 2]. It is hoped that findings of this study would guide policy and management decisions on how to improve the welfare of health workers by providing empirical data on the socio-demographic and job-related factors that influenced their experience of burnout during the COVID-19 pandemic so as to aid in preparing hospital staff against future pandemic-induced job-related burnout.

## Aim

This study sought to determine personal and job-related characteristics of health workers in Accra, Ghana that influenced their experience of burnout during the COVID-19 pandemic.

## Methods

### Study design

This study employed a cross-sectional survey design in which quantitative data was collected from health workers in three different hospitals in Accra. The hospitals were chosen to represent the primary, secondary and tertiary care facilities in Accra as per the classification of the Ministry of Health, Ghana [2, 6].

### Study setting and population

The study was conducted in Accra in the Greater Accra Region of Ghana. Accra is the capital and largest city of Ghana with a total population of 4,010,054 consisting of urban and rural populations of 3,630,955 and 379,099 respectively [8, 17, 18]. The high commercial activities with its associated high cost of living place huge demands on the residents of Accra [17]. In

addition, workers in both public and private institutions in Accra spend several hours to and from work due to heavy vehicular traffic on the roads of Accra [8, 19].

In this study, three public hospitals were purposively chosen to represent the three levels of the public healthcare system of Accra; primary, secondary and tertiary.

The Weija-Gbawe Municipal Hospital (WGMH) which was previously known as the Ga South Municipal Hospital or "Alkawe Hospital" was selected to represent the primary level of health care [20]. The primary level hospital was chosen because of the fact that it had been ranked as the best performing hospital among its peers using a peer ranking system established by the Ghana Health Service (GHS) [20]. The hospital had five hundred and seventy-six (576) health workers consisting of three hundred and eighty (380) clinicians and one hundred and ninety-six (196) non-clinicians. The Weija-Gbawe Municipal Hospital is located in the Weija-Gbawe Municipality in the Greater Accra Region of Ghana with an average daily admission ranging from sixty to one hundred (60–100) patients with over three hundred (300) daily patient attendance at the out-patient departments (OPDs) of the facility.

The Greater Accra Regional Hospital (GARH) also known as the Ridge hospital was selected to represent the secondary level of the health care system in Accra. The Ridge hospital receives referral cases from the district and sub-district health facilities and is the main secondary level health facility in Accra providing specialized care [20] and serves as the regional hospital for the Greater Accra Region [20]. The hospital had nine hundred and thirty-six (936) clinical and four hundred and sixty four (464) non-clinical staff. The Greater Accra Regional Hospital has an average daily admission ranging from one hundred to two hundred (100–150) patients with over eight hundred (800) daily patient attendance at the out-patient departments (OPDs) of the facility.

The Korle Bu Teaching Hospital (KBTH) was selected to represent the tertiary level of health care in Accra. The hospital is the national referral hospital of Ghana and the teaching hospital of the Greater Accra Region [19] and is currently the largest referral facility in the West African sub-region and the third largest hospital in Africa with a staff strength of four thousand, nine hundred and sixty-nine (4,969) consisting of three thousand one hundred and seventy eight (3,178) clinicians and one thousand seven hundred and ninety one (1,791) non-clinicians [19]. The hospital is known to receive huge number of referral cases from across the country and to have an average daily patient attendance at the out patient departments (OPDs) of almost two thousand (2000) with daily admission rate at almost three hundred (300) patients [19].

## Sample size determination

The sample size for the study was determined using the sample size determination table designed by Krejcie and Morgan [21]. This sample size determination method was chosen to ensure maximum sample size was obtained as the P = the population proportion for the outcome in all categories of the study population (was assumed to be .50 since this provided the maximum sample size) [21]. Krejcie & Morgan's table is a simple method for calculating sample size when the variance of the community and the percentage of error are not known [21]. Thus, use of Krejcie and Morgan to determine sample size is appropriate when the objective is to estimate population percentages of an outcome variable from sample percentages of the same outcome variable [21].

The total population of eligible clinical staff was three thousand six hundred and ninety (3,690) based on the eligible staff distributions for the three hospitals; 280, 836 and 2574 for primary, secondary and tertiary level hospitals respectively as obtained from the human resources (HR) unit of the three hospitals. Using the Krejcie & Morgan [21] tables for sample

size determination, these populations of eligible clinical staff corresponded to sample sizes of 162, 292 and 335 participants for the primary, secondary and tertiary level hospitals respectively. After adding 10% for non-response to the sample sizes, these corresponded to the adjusted sample sizes of 179, 322 and 369 participants for the primary, secondary and tertiary level hospitals respectively. Thus, sample size of 870 clinical staff was recruited for the study.

For the non-clinical staff, the total eligible population was one thousand nine hundred and fifty seven (1,957) based on the eligible staff distribution for the three hospitals which was 140, 277 and 1,540 for primary, secondary and tertiary level hospitals respectively. Using the Krejcie & Morgan [21] table for sample size determination, these populations corresponded to sample sizes of 103, 162 and 310 participants for the primary, secondary and tertiary level hospitals respectively. After adding 10% for non-response to the sample sizes, these corresponded to the adjusted sample size of 114, 178 and 341 participants for the primary, secondary and tertiary level hospitals respectively. Thus, a total sample size of 633 non-clinical staff was recruited for the study.

## Sampling technique

We included both clinical and non-clinical health workers in the three chosen hospitals in Accra. Health workers with not less than one year working experience were included for sampling while those on national service, contract staff and or staff on clinical rotation were excluded. Proportionate stratified sampling technique was used to select the participants from the accessible population at the three chosen health facilities. In selecting the participants, sampling proportionate to size was used to determine the number of health workers to be interviewed for both clinical and non-clinical categories and from the three facilities.

The individual participants were selected by simple random selection technique of balloting without replacement [22] by the researchers from the samples of clinicians and non-clinicians. Procedurally, a list of all eligible clinicians and non-clinicians were obtained from the HR units of the three hospitals. The eligible staff were put into strata (each professional group of clinicians and each rank of non-clinicians). All the eligible participants in each stratum (each professional group of clinicians and each rank of non-clinicians) were given unique codes. These unique codes were written on small pieces of papers and the pieces of paper placed into covered containers for each professional group of clinicians and each rank of non-clinicians and the covered containers were then shaken thoroughly. After which, the small pieces of papers were picked out at random from each covered container. The number pieces of paper picked from each container representing each stratum (each professional group of clinicians and each rank of non-clinicians) was determined based on staff distribution for the Greater Accra Region as per the Ghana Health Service (GHS) [20]; nurses/midwives (69%), doctors (16.1%), pharmacists/pharmacy technicians/ dispensing assistants (4.4%), biomedical scientists/laboratory assistants (3.9%) and the rest of the clinical grades, that is, audiologists, radiographers, dental clinical assistants, physiotherapists (6.6%) as well as senior staff of 38% and junior staff of 62% for non-clinicians [20]. The staff whose identities corresponded with the codes on the pieces of papers that were picked randomly from each container for each professional group and each rank of non-clinical staff were contacted for written informed consent and recruitment at their various units of work in their various hospitals.

During the study measures aimed at preventing COVID-19 infection such as; maintaining physical distancing, wearing of nose mask and facial shields, use of alcohol-based hand-sanitizers among others as recommended by the Ghana Health Service [2] were strictly adhered-to during the data collection. The data collection took place between March to November, 2020.

## Data collection

Once selected and written informed consent obtained, the participants were given the study questionnaire by trained research assistants (RA), who had at least a Bachelor of Science in nursing, at the rest rooms or conference rooms of their various hospitals. The role of each participant was to complete and return the study questionnaire in a sealed brown envelope (which was provided by the investigators) within twenty four hours after they had provided written informed consent. Each participant spent between thirty to forty five minutes to complete the study questionnaire. In line with the guidelines of the GHS against the COVID-19 pandemic, all the RAs and participants were given a free re-usable face mask at no cost to them to wear during the period of the data collection. Also, "Veronica buckets" (a localized equipment for hand washing in Ghana) with water and soap were provided for hand washing with soap under running water in facilities where there was no access to flowing water. Furthermore, alcohol-based hand sanitizers were provided free of charge for the RAs and participants to use in order to protect them from the deadly COVID-19 virus. Additionally, social distancing (6 feet between people) was maintained as far as possible throughout data collection. No other interventions or treatment were administered to any of the participants and no manipulation of the respondents or study environment were done. The participants kept a copy of the signed written informed consent form.

## Instrument

Data was collected using a pre-tested questionnaire which was self-administered. It took each participant between thirty to forty five minutes to complete the questionnaire even though they were required to submit the filled out questionnaire within twenty four hours. The questionnaire contained questions which collected socio-demographic and job-profile information. Also, the questionnaire assessed participants' level of resilience and burnout syndrome. Information on the questionnaire was explained to the study participants to ensure they filled it out without difficulties. Each participant completed and returned the study questionnaire in a sealed brown envelope (which was provided by the researchers) within twenty four (24) hours. Each questionnaire was given a unique code in order to ensure confidentiality of the responses.

## Level of resilience

In other to determine the level of resilience, the Brief Resilience Scale (BRS) was used [23, 24]. The BRS is a six-item (statement) scale. The participants were asked to indicate how well each statement described their behaviour and actions on a 5-point likert-type scale, ranging from "1" = does not describe me at all to "5" = describes me very well. Item 2 (I have a hard time making it through stressful events), Item 4 (It is hard for me to snap back when something bad happens) and Item 6 (I tend to take a long time to get over set-backs in my life) were reverse-coded and this was taken into consideration during the data collection, entry and analysis. The BRS was scored by adding the responses varying from 1–5 for all six items giving a range from 6–30. The total scores were then divided by the total number of questions answered to give the score corresponding to the level of resilience as recommended in literature [24] Participant scores on the BRS were categorized into 1.0 to 2.99, 3.0 to 4.30 and 4.31 to 5.0 and these scores corresponded to low resilience, normal resilience and high resilience respectively [24]. The original six-item structure of the BRS was maintained for this study (Cronbach's α = 0.78) [24].

## Burnout syndrome

Burnout syndrome was assessed using an adapted 22-item Maslach Burnout Inventory–Human Services Survey tool (MBI-HSS) [3, 23, 25]. The inventory has been widely used and has an overall Cronbach's Alpha of 0.87 [25]. Each inventory item was rated on a seven-point Likert scale that measures how frequently the participants experience a particular feeling (from 0 for never to 6 for every day). The MBI-HSS tool measures the three constructs of burnout: Emotional Exhaustion (EE), using nine (9) items to measure physical and emotional depletion; Depersonalization (DP), using five (5) items to measure negative or cynical feelings about patients; and Personal Accomplishment (PA), using eight (8) items to measure how one perceives one's own competence.

In this study, some of the questions of the original MBI-HSS tool (questions 11 and 20) were modified in a pilot to make it more appropriate for use in Ghana (Cronbach's alpha scores: 0.94 for EE, 0.83 for DP, and 0.86 for PA). The adapted MBI-HSS was used to collect the final information on burnout syndrome among the participants.

The scores on the EE dimension subscale were categorized Low EE<16, Moderate EE = 17–26 and High EE > 27. Also, the scores on the DP dimension were categorized as Low DP<6, Moderate DP = 7–12 and High DP > 13. In addition, the scores on the PA subscale was categorized into Low PA <31, Moderate PA = 32–38 and High PA>39.

In this study, we adopted the definition of burnout as provided in literature [3, 23, 26] as participants with high scores on EE and DP and low scores on PA [3, 23, 26]. Thus, participants who had high EE, high DP and low PA were said to be experiencing burnout while those who did not fall in this category were classified as not having burnout.

## Data analysis

The data was initially entered into Microsoft Excel 2016 and imported to Stata 15.0 to aid with the analysis. A p-value of less than 0.05 was deemed to be significant. Descriptive statistics were used to present the proportions on socio-demographic and job profile of the participants at the univariable level. Overall internal consistency of the 22-item adapted MBI-HSS tool was determined using Cronbach's alpha estimation. Bivariable models were used to predict the associations between burnout and socio-demographic as well as job-related factors. Pearson's chi-square test of independence was performed to test for individual independent association between the burnout and socio-demographic as well as job-related with observations more than six. However, for variables with observations less than six, which were also nominal and dichotomized a Fisher's exact test, were used to determine the associations.

After the binary analysis, multivariable logistic regression was used to determine the strength of association. The multivariable logistic regression was done at two levels. The first was to establish the crude level of association between the socio-demographic as well as job-related factors and burnout. In this strategy, all independent variables were added to the first model to establish the crude relationship with the dependent variable at various levels of measurement. Afterwards, variables that were significant in the first model and at the bivariable analysis level were put in a multivariable logistic regression model to adjust for the effects of those variables.

## Ethical approval and consent to participate

The data in this paper is part of a study titled "Burnout, metabolic syndrome and predisposition to non-communicable diseases: A study among health workers in the Greater Accra Region of Ghana" in which ethical approval was obtained from the Ghana Health Service's Ethics Review Committee (Protocol Number: GHS-ERC 018/03/20). Similar approval was

obtained from the Institutional Review Board of the Korle-Bu Teaching Hospital (Protocol Number: KBTH-IRB /000133/2018). No publication has been made from the study yet. Permission was also sought from the management of the three hospitals before data collection. Written informed consent was also obtained from each participant before recruitment into the study. Each participant was informed of his/her right to withdraw from the study at any time without suffering any negative consequences. Names of the participants were not revealed in the study report and all information gathered from the study participants were treated confidentially as all questionnaires were placed in a special cabinet under lock and only accessible by the researchers only. All the COVID-19 prevention protocols were adhered-to during the data collection.

## Results

### Socio-demographic and job-profiles of participants

The total number of participants who took part in the study was one thousand two hundred and sixty four health workers (1,264) and this represented a response rate of 84.10%. The participants' mean age was 40.81 ± 8.33 years with the participants having their ages ranging between twenty four to fifty six (24–56) years. 40.78% of the participants were within the age brackets of 29–39 years. More than half of the participants (53.09%) were females. Also, majority (61.31%) of the participants were married. Only 36.08% of the participants claimed to have children. In addition, majority (65.59%) of the participants claimed to have obtained tertiary education. Three hundred and twenty six (326) (25.79%) of the participants had low level of resilience. In addition, 42.32% of the participants had a working history of 1–5 years with 45.49% rendering their services in a tertiary facility. More than half (55.62%) of the participants were clinicians (S1 Table provides information on the comparison of clinicians and non-clinicians on the socio-demographic and job-related factors). One hundred and seventy eight (178) (14.08%) of the participants claimed to be rendering their services in highly dependent units (Emergency and Intensive Care Units). Also, 36.71% of the participants rendered their services in stable-in patients. Night shifts which was defined as duty times between 8pm to 8am Ghana time as compared to afternoon shifts defined as duty times between 1:30pm to 8:00pm and morning shifts defined as duty times between 7:30am to 2:00pm were the predominant shift engaged by the participants for the past six months. Five hundred and forty four (544) (43.04%) of the participants said they had additional jobs aside their current jobs with half of the participants (50%) harboring intentions of leaving their current jobs. 40.11% of the participants claimed they had adequate support for their jobs as defined by adequate support from colleagues and management with 46.20% of the participants claiming to have control over their jobs defined as being autonomous on the job and being able to make decisions on the job. Furthermore, majority of the participants said they were not satisfied with their jobs with 20.57% perceiving their current jobs as imposing high workload on them (Table 1).

### Mean, standard deviation and internal consistency of adapted MBI-HSS

The results showed that the overall reliability of the scores on the adapted MBI-HSS tool (items 1–22) was sufficient (Cronbach's $\alpha$ = 0.88), with the lowest internal consistency value for the scale of depersonalization (DP) (Cronbach's $\alpha$ = 0.83).

### Distribution of elements of burnout syndrome among the participants

The results showed that there was no significant difference in the rate of emotional exhaustion and personal accomplishment between clinicians and non-clinicians. However, non-clinicians

**Table 1. Socio-demographic and job-profiles of participants.**

| Socio-demographic and job-related factors | Frequency | Percentage |
|---|---|---|
| **Age** | | |
| 18–28 | 78 | 5.54 |
| 29–39 | 515 | 40.78 |
| 40–50 | 447 | 35.39 |
| 51–61 | 231 | 18.29 |
| **Sex** | | |
| Male | 593 | 46.91 |
| Female | 671 | 53.09 |
| **Marital status** | | |
| Single | 205 | 16.22 |
| Married | 775 | 61.31 |
| Divorced | 150 | 11.87 |
| Co-habitation | 72 | 5.70 |
| Separated | 62 | 4.91 |
| **Having a child** | | |
| Yes | 456 | 36.00 |
| No | 808 | 63.92 |
| **Highest Educ. Qualification** | | |
| Low | 33 | 2.59 |
| Middle | 322 | 25.24 |
| High | 921 | 72.18 |
| **Years of working** | | |
| 1–5 | 565 | 42.32 |
| 6–10 | 221 | 16.55 |
| 11–15 | 73 | 5.47 |
| 16–20 | 104 | 7.79 |
| ≥20 years | 372 | 27.87 |
| **Net monthly income** | | |
| Low | 734 | 57.84 |
| Middle | 470 | 37.04 |
| High | 65 | 5.12 |
| **Level of Resilience** | | |
| Low | 326 | 25.69 |
| Normal | 483 | 38.06 |
| High | 460 | 36.25 |
| **Facility of working** | | |
| Primary | 283 | 22.39 |
| Secondary | 406 | 32.12 |
| Tertiary | 575 | 45.49 |
| **Category of staff** | | |
| Clinicians | 703 | 55.62 |
| Non-clinicians | 561 | 44.38 |
| **Service Area** | | |
| Highly dependent unit | 176 | 13.92 |
| Stable in-patients | 464 | 36.71 |
| Out-patient | 264 | 20.89 |
| No contact with patients | 304 | 24.05 |

(*Continued*)

**Table 1.** (Continued)

| Socio-demographic and job-related factors | Frequency | Percentage |
|---|---|---|
| Management | 56 | 4.43 |
| **Common shift for past 6 months** | | |
| Night | 601 | 47.55 |
| Afternoon | 274 | 29.59 |
| Morning | 289 | 22.86 |
| **Additional Job** | | |
| Yes | 544 | 43.04 |
| No | 720 | 56.96 |
| **Intention to leave job** | | |
| Yes | 632 | 50.00 |
| No | 632 | 50.00 |
| **Job support** | | |
| Yes | 507 | 40.11 |
| No | 757 | 59.89 |
| **Job control** | | |
| Yes | 584 | 53.80 |
| No | 680 | 46.20 |
| **Job satisfaction** | | |
| Yes | 545 | 43.12 |
| No | 719 | 56.88 |
| **Perceived high workload** | | |
| Yes | 632 | 50.00 |
| No | 632 | 50.00 |

displayed higher depersonalization. The results showed that participants from the primary level hospital had the highest level of high emotional exhaustion and depersonalization. In contrast, the level of low personal accomplishment was highest in the tertiary level hospital (Table 2).

## Prevalence of burnout among participants

The percentage of participants with burnout as defined as participants with the three defining characteristics of burnout; high EE, high DP and low PA was 260 (20.57%). Compared to the

**Table 2. Distribution of elements of burnout syndrome among the participants.**

| | EE | | | | DP | | | | PA | | | |
|---|---|---|---|---|---|---|---|---|---|---|---|---|
| | Low | Mod. | High | p value | Low | Mod. | High | P value | Low | Mod. | High | p value |
| **CATEGORY OF STAFF** | | | | **0.172** | | | | **0.004** | | | | **0.220** |
| Clinical | 280 | 199 | 224 | | 515 | 38 | 150 | | 283 | 115 | 305 | |
| Non-clinical | 195 | 168 | 198 | | 375 | 22 | 164 | | 199 | 99 | 263 | |
| **FACILITY** | | | | **<0.001** | | | | **<0.001** | | | | **<0.001** |
| Primary | 115 | 42 | 126 | | 169 | 12 | 102 | | 99 | 72 | 112 | |
| Secondary | 148 | 127 | 131 | | 288 | 21 | 97 | | 165 | 63 | 178 | |
| Tertiary | 212 | 198 | 165 | | 433 | 27 | 115 | | 218 | 79 | 278 | |

Data are presented as frequency (n) and p-values were determined using $\chi 2$.

**Table 3. Prevalence of burnout among participants.**

| Profile of participants | Burnout | |
|---|---|---|
| | Frequency (p) | p- value |
| **Category of staff** | | < 0.001 |
| Clinicians | 110 (15.65) | |
| Non-clinicians | 150 (26.74) | |
| Total | 260 (20.57) | |
| **Facility of working** | | < 0.001 |
| Primary | 102 (36.04) | |
| Secondary | 69 (17.12) | |
| Tertiary | 89 (15.48) | |

Data are presented as frequency (percentages) and p-values were determined using χ2.

clinical participants, the non-clinical participants had a significantly higher prevalence of burnout (26.74% versus 15.64%, p < 0.001). Also, participants at the primary level of health-care had the highest prevalence of burnout (36.04%) (Table 3).

## Bivariate association between burnout and socio-demographic factors

A bivariable analysis was conducted to determine the socio-demographic and job-related factors associated with burnout among the participants. The results showed that beside the sex of participants, all the other socio-demographics (age, marital status, having a child, highest education, years of working experience, net monthly income, and level of resilience) were significantly associated with burnout. Majority of the burnt-out participants (53.08%) were within the age brackets of 29–39 years. Also, 72.31% of those burnt-out were married. Further, 81.54% of those who were burnt-out had children. In addition, more than half (56.92%) of those burnt had significantly higher education. Burnout was found to be significantly high among participants with less years of working experience (1–5 years) (70.77%). 65.38% of participants who were burnt-out were staff with low net monthly income. Low level of resilience was found to be significantly associated with burnout; 87.31% of the participants who were burnt had low level of resilience (Table 4).

Also, the results showed that aside job support and job control, all the other job-related factors (facility of working, category of staff, service area, common shift for the past six months, additional jobs, harboring of intentions to leave job, job dissatisfaction and perceived high workload) were significantly associated with burnout. More than half (57.69%) of those burnt were non-clinicians. 31.92% of those who were burnt rendered their services in highly dependent units. Also, night shifts were associated with significantly higher level of burnout. Further, burnout was found to be significantly associated with additional jobs. In addition, participants who expressed intentions of leaving their jobs were more burnt. Similarly, burnout was found to be significantly associated with perceived high workload and job dissatisfaction (Table 4).

## Multivariate logistic regression between burnout and socio-demographic and job-related factors

In order to determine the socio-demographic factors that significantly influence the experience of burnout among health workers, binary logistic regression analysis was carried out using two models (Table 5). Model I was a bivariate logistic regression analysis of burnout and all the socio-demographic factors that had an association with burnout in Table 4. Model II

**Table 4. Bivariate analysis of burnout and socio-demographic and job-related factors.**

| Socio-demographic and job-related factors | Burnout N (%) 260 (100) | χ2, p-value |
|---|---|---|
| **Age** | | **123.5, < 0.001** |
| 18–28 | 39 (15) | |
| 29–39 | 138 (53.08) | |
| 40–50 | 79 (30.38) | |
| 51–61 | 4 (1.54) | |
| **Sex** | | 0.482, 0.488 |
| Male | 117 (45.00) | |
| Female | 143 (55.00) | |
| **Marital status** | | **72.48, < 0.001** |
| Single | 63 (24.23) | |
| Married | 188 (72.31) | |
| Divorced | 7 (2.69) | |
| Co-habitation | 1 (0.38) | |
| Separated | 1 (0.38) | |
| **Having a child** | | **293.4, <0.001** |
| Yes | 212 (81.54) | |
| No | 48 (18.46) | |
| **Highest Educ. Qualification** | | **46.75, <0.001** |
| Low | 6 (2.31) | |
| Middle | 106 (40.77) | |
| High | 148 (56.92) | |
| **Years of working** | | |
| 1–5 | 184 (70.77) | **215.1, < 0.001** |
| 6–10 | 65 (25.00) | |
| 11–15 | 4 (1.54) | |
| 16–20 | 4 (1.54) | |
| ≥20 years | 2 (0.77) | |
| **Net monthly income** | | **26.07, <0.001** |
| Low | 170 (65.38) | |
| Middle | 67 (25.77) | |
| High | 23 (8.85) | |
| **Level of Resilience** | | **649.5, < 0.001** |
| Low | 227 (87.31) | |
| Normal | 26 (10.00) | |
| High | 7 (2.69) | |
| **Facility of working** | | 53.77, < 0.001 |
| Primary | 102 (39.23) | |
| Secondary | 69 (26.54) | |
| Tertiary | 89 (34.23) | |
| **Category of staff** | | |
| Clinicians | 110 (42.31) | 23.49, < 0.001 |
| Non-clinicians | 150 (57.69) | |
| **Service Area** | | 176.7, <0.001 |
| Highly dependent unit | 83 (31.92) | |
| Stable in-patients | 82 (31.54) | |
| Out-patient | 3 (1.15) | |
| No contact with patients | 70 (26.92) | |

(*Continued*)

**Table 4.** (Continued)

| Socio-demographic and job-related factors | Burnout N (%) 260 (100) | χ2, p-value |
|---|---|---|
| Management | 12 (4.62) | |
| **Common shift for past 6 months** | | 13.46, 0.001 |
| Night | 101 (38.85) | |
| Afternoon | 86 (33.08) | |
| Morning | 79 (30.38) | |
| **Additional Job** | | 97.06, <0.001 |
| Yes | 182 (70.00) | |
| No | 78 (30.00) | |
| **Intention to leave job** | | 217.6, <0.001 |
| Yes | 236 (90.77) | |
| No | 24 (9.23) | |
| **Job support** | | 2.134, < 0.144 |
| Yes | 94 (36.15) | |
| No | 166 (63.85) | |
| **Job control** | | 0.920, 0.337 |
| Yes | 127 (48.85) | |
| No | 133 (51.15) | |
| **Job satisfaction** | | 4.381, 0.036 |
| Yes | 127 (48.85) | |
| No | 133 (51.15) | |
| **Perceived high workload** | | 217.6, <0.001 |
| Yes | 236 (90.77) | |
| No | 24 (9.23) | |

Data are presented as frequency (percentages) and p-values were determined using χ2.

was a multivariate logistic regression analysis of burnout and all the socio-demographic characteristics adjusting for level of resilience and job satisfaction among the participants. Health workers with a child were 0.11 times less likely (AOR = 0.11, 95% CI = 0.07–0.17) to experience burnout. Also, staff with 1–5 years working experience were 26.81 (AOR = 26.81, CI = 6.37–112.9) more likely to experience burnout. Similarly, participants within the 6–10 years of working experience were 14.60 times more likely (AOR = 14.60, 95% CI = 3.32–64.25) to experience burnout. In addition, participants with 11–15 years of working experience were 6.60 times more likely (AOR- = 6.60, 95% CI = 1.07–40.76) to experience burnout on the job. Further, 16–20 years of working experience among the participants were 5.55 times more likely (AOR- = 5.55, 95% CI = 0.94–32.97) to experience burnout on the job (Table 5).

Likewise, in order to determine the influence of the job-related factors on burnout among the participants, binary logistic regression analysis was also carried out using two models (Table 5). Model I was a bivariate logistic regression analysis of burnout and all the job-related factors that had an association with burnout in Table 4. Model II was a multivariate logistic regression analysis of burnout and all the job-related characteristics adjusting for level of resilience and job satisfaction. It was revealed that employees of the primary level facility were 3.91 times more likely (AOR = 3.91, 95% CI = 2.39–6.41) to experience burnout on the job. Also, non-clinicians were 2.57 times more likely (AOR = 2.57, 95% CI = 1.73–3.83) to experience burnout. Similarly, participants with additional jobs were 1.14 times more likely (AOR = 1.14, 95% CI = 0.75–1.74) to experience burnout. In addition, participants harbouring intentions of

**Table 5. Multivariate logistic regression between burnout and socio-demographic and job-related factors.**

| Socio-demographic and job-related factors | Burnout | | | | | |
|---|---|---|---|---|---|---|
| | Crude Odd ratio | 95% CI | P-value | Adjusted Odd ratio | 95% CI | P-value |
| **Age** | | | | | | |
| 18–28 | Reference | | | | | |
| 29–39 | 0.290 | 0.17–0.48 | < 0.001 | 0.54 | 0.27–1.10 | 0.090 |
| 40–50 | 0.17 | 0.10–0.29 | < 0.001 | 0.52 | 0.25–1.09 | 0.084 |
| 51–61 | 0.01 | 0.00–0.04 | < 0.001 | 0.06 | 0.02–0.26 | **<0.001** |
| **Marital status** | | | | | | |
| Single | Reference | | | | | |
| Married | 0.72 | 0.51–1.01 | 0.060 | 0.89 | 0.53–1.47 | 0.646 |
| Divorced | 0.11 | 0.05–0.25 | <0.001 | 0.09 | 0.04–0.244 | **< 0.001** |
| Co-habitation | 0.03 | 0.00–0.23 | 0.001 | 0.04 | 0.00–0.30 | **0.002** |
| Separated | 0.04 | 0.00–0.27 | 0.001 | 0.11 | 0.01–0.90 | **0.040** |
| **Having a child** | | | | | | |
| No | Reference | | | | | |
| Yes | 0.07 | 0.05–0.10 | < 0.001 | 0.11 | 0.07–0.17 | **< 0.001** |
| **Highest Educ. Qualification** | | | | | | |
| Low | Reference | | | | | |
| Middle | 0.86 | 0.35–2.12 | 0.069 | 0.58 | 0.16–2.14 | 0.583 |
| High | 2.34 | 0.94–5.84 | 0.746 | 2.07 | 0.55–7.86 | 0.285 |
| **Years of working** | | | | | | |
| 1–5 | 110.2 | 27.12–447.4 | < 0.001 | 26.81 | 6.37–112.9 | **< 0.001** |
| 6–10 | 77.08 | 18.64–318.7 | < 0.001 | 14.60 | 3.32–64.25 | **< 0.001** |
| 11–15 | 10.72 | 1.93–59.70 | 0.007 | 6.60 | 1.07–40.76 | **0.042** |
| 16–20 | 7.40 | 1.34–40.98 | 0.022 | 5.55 | 0.94–32.97 | 0.059 |
| ≥20 years | Reference | | | | | |
| **Net Month Income** | | | | | | |
| High | Reference | | | | | |
| Middle | 0.27 | 0.14–0.48 | <0.001 | 0.19 | 0.08–0.45 | **<0.001** |
| Low | 0.48 | 0.28–0.84 | 0.010 | 0.55 | 0.23–1.30 | 0.174 |
| **Facility of working** | | | | | | |
| Primary | 3.08 | 2.20–4.29 | <0.001 | 3.91 | 2.39–6.41 | < 0.001 |
| Secondary | 1.12 | 0.79–1.58 | 0.525 | 1.11 | 0.70–1.75 | 0.665 |
| Tertiary | Reference | | | | | |
| **Category of staff** | | | | | | |
| Clinicians | **Reference** | | | | | |
| Non- clinicians | 1.97 | 1.49–2.59 | <0.001 | 2.57 | 1.73–3.83 | < 0.001 |
| **Service Area** | | | | | | |
| Highly dependent unit | Reference | | | | | |
| Stable in-patients | 0.19 | 0.13–0.28 | < 0.001 | 0.21 | 0.12–0.38 | <0.001 |
| Out-patient | 0.01 | 0.000.03 | < 0.001 | 0.01 | 0.00–0.05 | <0.001 |
| No contact with patients | 0.27 | 0.18–0.40 | < 0.001 | 0.37 | 0.20–0.68 | 0.001 |
| Management | 0.24 | 0.12–0.49 | < 0.001 | 0.16 | 0.06–0.41 | <0.001 |
| **Common shift for past 6 months** | | | | | | |
| Morning | Reference | | | | | |
| Afternoon | 1.35 | 0.97–1.87 | 0.074 | 0.94 | 0.60–1.49 | 0.805 |
| Night | 1.86 | 1.33–2.61 | < 0.001 | 1.10 | 0.68–1.78 | 0.689 |
| **Additional Job** | | | | | | |

*(Continued)*

**Table 5.** (Continued)

| Socio-demographic and job-related factors | Burnout | | | | | |
| --- | --- | --- | --- | --- | --- | --- |
| | Crude Odd ratio | 95% CI | P-value | Adjusted Odd ratio | 95% CI | P-value |
| No | Reference | | | | | |
| Yes | 4.14 | 3.08–5.56 | < 0.001 | 1.14 | 0.75–1.74 | 0.042 |
| **Intentions to leave job** | | | | | | |
| No | Reference | | | | | |
| Yes | 15.10 | 9.74–23.41 | < 0.001 | 4.61 | 2.73–7.78 | < 0.001 |
| **Perceived high workload** | | | | | | |
| No | Reference | | | | | |
| Yes | 15.10.. | 9.73–23.40 | < 0.001 | 4.60 | 2.73–7.77 | < 0.001 |

Adjusted for level of resilience and job satisfaction.

leaving their current jobs were 4.61 times more likely (AOR = 4.61, 95% CI = 2.73–7.78) to experience burnt-out on the job. Furthermore, participants with perceived high workload were 2.38 times more likely (AOR = 2.38, 95% CI = 1.40–4.05) to experience burnt-out on the job (Table 5).

## Discussion

More than half of the participants in this study were females. This is similar to what previous studies have reported that females constitute the majority of the health workforce in SSA [1, 4]. The high number of females among the participants in this study was because nurses and midwives who form majority of the health workforce are predominantly females [5, 9]. The study further found that almost 26% of the participants reported with low level of resilience. Low level of resilience predisposes health workers to psycho-social disorders such as burnout [8, 26] as it reduces their coping abilities to stress [10, 12]. Afulani et al. [2] have suggested that increasing the level of resilience of health workers is one of the effective ways of reducing burn-out occurrence and this recommendation is supported by earlier studies [11, 26].

In addition, the results revealed that majority of the health workers were clinicians and this is supportive of studies in Ghana [8, 16] that have established clinicians as the dominant category of hospital staff. Also, 43.04% of the participants reportedly did additional jobs aside their main hospital jobs with half (50%) of the participants harboring intentions of leaving their current hospital jobs. This finding of high intentions of wanting to leave the job is in line with previous studies that found that majority of health workers in Egypt, Ghana and Nigeria harbored intentions of leaving their current jobs for high rewarding jobs in other jurisdictions [1, 4, 23]. Majority of healthcare workforce in SSA contemplate migrating to other countries where they perceived healthcare work to be less stressful but more rewarding in terms of income [1–4]. Also, the high intentions of wanting to leave current jobs could be as a result of the high workload and its associated burnout [1, 8]. Burnout is associated with apathy towards work [2, 4] and this could trigger intentions of wanting to quit job [23, 24, 26–28].

The results revealed that over a quarter of the health workers were experiencing burnout. This is in line with the findings of [4] that there was high burnout among health workers in SSA. Afulani et al. [2] reported that burnout was high among Ghanaian health workers and that inadequate preparedness towards the COVID-19 pandemic further compounded the burnout situation even among health workers not in contact with COVID -19 patients. Also, the findings are in line with those found by [12] where almost 30% of health workers were found to be burnt. Similarly, He et al. [27] reported that burnout was high among health

workers in China. However, the prevalence of burnout in this study is higher than the prevalence found by [8] among some sections of Ghanaian health workers using Pine's burnout tool. Similarly, the findings are in contrast with those found by Opoku and Appenteng [13], among physicians in Ghana. Furthermore, the findings are also in contrast with the findings of [28] who observed that burnout rates were lower among health workers in India.

Regarding the distribution of the three elements of burnout syndrome, the results showed that 33.39% of the participants had high level of emotional exhaustion and were said to be emotionally burnt. This finding is consistent with those found by [4]. Similarly [11], found high level of emotional exhaustion among nurses in their study. This finding is however in contrast to those found by [9] where it was found that emotional exhaustion was generally low among hospital staff. Similarly, Opoku and Apenteng [13], found emotional exhaustion to be lower among physicians in Ghana. In addition, 25.31% of the health workers had depersonalization and similar findings were found by He et al. [27] and Afulani et al. [2]. However [10], found contrary findings (16.6%). Also, 37.87% of the participants had low personal accomplishment as compared to 20% found by [28]. However, similar rates (39.72%) were found by [4]. This finding of high reduced personal accomplishment could be responsible for the increased intentions to leave their current jobs as they no-longer find their jobs to be exciting enough [2]. Similar finding has been reported recently [27].

This finding of high burnout rate could be because of the reduced resources for health care particularly with the emergence of COVID-19 [2, 14] as well as the associated increased workload following the COVID-19 pandemic and confounding role of the fear of contracting the deadly COVID-19 in the line of work as was stated by [2, 11] as compared to the earlier Ghanaian studies [8, 13–16]. Inadequate preparedness of health workers to handle emerging diseases such as COVID-19 [11, 14], HIV/AIDS [5], and cancers [3] have been cited as contributory to the rising rates of burnout among health workers. Also, the high burnout reported in this study could be because of use of a representative and diverse sample in this study and hence provided a clearer perspective of the burnout situation in Ghanaian health facilities as compared to the earlier Ghanaian studies [8, 13] that used non-representative samples.

Furthermore, the relatively high rates of burnout found in this study can be attributed to the numerous job-related [1, 5] and personal [2–5] challenges experienced by health workers. The under-resourced health care system [9–12] coupled with inadequate hospital staff [1, 8] puts high demands on the few health workers in SSA and these could trigger burnout. Finally, the absence of a structured mentorship system in Ghanaian hospitals [15, 16] to help new employees cope with work and develop high resilience has been suggested to be implicated in the rising burnout as well as exodus of health workers to other jurisdictions [1].

Contrary to previous studies [3, 7] which have found that female healthcare providers tend to have higher burnout, this study did not find any significant relationship between sex of the health worker and burnout. This is in line with some studies [11–13] that have concluded that gender does not influence burnout development. However, several divergent studies have showed female health workers have more vulnerability to emotional exhaustion than males and as a result were more prone to burnouts than their male counterparts [24, 26]. Furthermore, some studies [4, 27] suggest that women tend to report higher emotional exhaustion scores; while men tend to report higher depersonalization and low personal accomplishment scores [5].

Also, the study found that burnout is more common among younger and inexperienced staff similar to what was found by other studies [9, 28]. On the contrary, Mbanga et al. [5] found that older health workers aged forty to fifty (40–50) years have a greater subjection to psychological and physical oppressions caused by stress resulting from overworking and

carrying out tedious duties and this resulted in burnout. The majority of studies [2–5] are inconclusive, as half of them found no burnout differences comparing young and senior health workers. The high burnout among the young health workers could largely be due to the low resilience of young health workers [7, 23].

Low resilience was identified as yet another risk factor for the development of burnout among the health workers and this is similar to the findings of [2, 29]. Health workers with low resilience tend to have poor coping abilities when exposed to stressors at the workplace and easily develop burnout compared to those with normal and high resilience [29]. Resilience building among health workers provides a plausible mechanism of reducing burnout [2, 29].

The study found that married health workers had higher rate of burnout as compared to singles and divorced. This finding is in contrast to the finding of Dubale et al. [3] who found that health workers prevented burnout mainly by support from family members and interests/ hobbies. Thus, married health workers often had low rate of burnout due to spousal support [29–32]. This observation stands to reason because families are sources of all kinds of support including moral, emotional, financial, and physical support.

Furthermore, the study found that high educational qualification and less years of work experience was associated with burnout. This is in line with the work of Odonkor and Frimpong [8], who found an association between burnout and these socio-demographic characteristics: age, gender, educational qualification, occupation, years of work experience, marital status, and parenthood (having children). However [13], in their work among physicians in Ghana stated that the association between burnout and age, sex, years of practice and clinical specialty were not statistically significant. The high burnout among those with high educational qualification could largely be due to the increased demands at work on highly educated staff and this could overwhelm them leading to burnout experience.

The results revealed that non-clinicians were found to have higher burnout and this could partly be due to the high level of resilience that was found among clinicians as compared to the non-clinicians. Resilience is one mechanism in which individuals use to adapt and live with stressful situations [2, 30]. Individuals with high level of resilience tend to have increased capacity to deal with stressors and therefore adapt better to stressful situations at the workplace [31, 32]. Also, the increased job demands on the non-clinicians as they interact with different cadre of staff; both clinical and non-clinical places increase tendency for them to have interpersonal conflicts and the likelihood of burnout [7–10].

Further, the results revealed that health workers at the primary level of hospital were more likely to have higher rate of burnout compared to their counterparts at the secondary and tertiary level hospitals. This finding could be because those at the primary level hospital had less supportive colleagues compared to those at the higher level of healthcare [2, 8]. Also, the low resource allocation to the primary level hospital [2, 6] could predispose workers at that level to higher rate of burnout since this often results in a mismatch between job demands and resources [2, 5]. However, this findings is in contrast to those found by [14, 23] who stated that working in specialist and tertiary facilities involved a lot of interactions between various specialist health workers and such interpersonal relationships could trigger burnout as compared to working at primary level hospital. Abdo and Kabbash [23], further assert in their study that the workload at the primary level of care was less demanding as compared to the secondary and tertiary levels which served as referral points for the numerous primary level facilities and this led to increased workload and its associated burnout.

Also, participants on night duties between 8am to 8pm Ghana time were found to have significantly higher level of burnout and this could be due to the low material [4, 7] and human [15, 27] resources during night shifts. Night shifts often have inadequate staff [1, 16] and this placed increased demands on the available staff with its associated burnout [2–5]. The low

resource allocation during night and weekend shifts put staff on these shifts to increased tendency for burnout [4, 23]. Furthermore, the sleep deprivation and associated hormonal imbalance among health workers doing night shifts makes them emotionally labile and thus at increased risk of burnout [28–31]. The relaxation effect of high melatonin level [29–32] associated with a good night sleep is minimally available for health workers engaged on night shifts and hence predispose them to burnout [23, 24, 26–31].

In addition, working in High Dependency Units (HDUs) such as the Intensive Care Units (ICUs) and/or Emergency Units was associated with burnout. Working in HDUs was associated with increased job demands on hospital staff and this predisposed hospital staff in these units to burnout [2, 30]. Similarly, Mbanga et al. [5] stated that working in emergency and maternal settings was associated with high burnout. Also [7, 26], found that burnout was more common among staff of ICUs. Staff engaged in HDUs ought to be given supportive supervision to prevent them from getting burnt as burnout could result in sub-optimal care such as medication errors and physical health conditions in such staff [10–12].

Participants who did additional jobs aside their current hospital jobs were more likely to get burnt because doing an additional job placed additional strain [8] on health workers and this predisposed such staff to increased risk of burnout [13]. This finding is similar to those found by [23] and also those found by [27]. Hospital staff engaged in additional jobs aside their primary jobs often known as "locum work" [13–16] tend to have less time to relax and are constantly under stress. This constant job strain makes affected health workers to become burnt [2, 27] and harbor intentions of leaving their current hospital jobs [1, 4]. In addition, the study further found that employees who perceived their current jobs as imposing high workload were more burnt. This finding is in line with those found by Dubale et al. [3, 10, 27] who stated that the perception and existence of high of workload influenced an employee's level of burnout.

The results further found that burnout was higher among those with no job satisfaction. Job dissatisfaction is a known determinant of burnout [3, 26, 32]. Individuals who feel dissatisfied with their jobs are more likely to be emotionally unstable and at risk of burnout [10]. This finding of job dissatisfaction being associated with high burnout is in line with the findings of [9–11].

## Limitations of the study

Firstly, the design that was used for the study was a cross-sectional design; hence, causality cannot be inferred from the findings. The study results only depict a relationship between burnout and the socio-demographic as well as the job-related factors and do not depict causation. However, the findings provide insights for possible longitudinal and experimental studies to establish possible causation.

The study was conducted in only three public hospitals out of the over eight hundred in the Greater Accra region of Ghana, and hence there may be a limit to the extent to which the findings can be generalized to other health workers in hospitals that were not selected. However, to overcome the effects of this limitation, the chosen hospitals were purposively selected to represent the three levels of the public health care system in Accra [1]. It is believed that the hospitals that were not selected shared common features with the selected facilities and that the health workers were invariably similar in characteristics [2]. However, the study used probability sampling method to select the participants from the accessible population and this makes the findings of the study generalizable to the accessible and target population. Also, the study involved different categories of health workers who were randomly selected and this contributed to the external validity of the study findings.

In addition, this study was susceptible to survivor bias because it assessed prevalence rather than incident cases. It is also likely that all the scores were subjected to reporting bias, since the data here were collected through questionnaire. To overcome this limitation, the participants were asked to frankly answer the questions on the questionnaire and to try and remember incidents before choosing their responses on the questionnaire.

Also, the high burnout in this study could also be influenced by other factors aside the COVID-19 pandemic such as the grief caused by multiple losses [11], personal characteristics of health workers which are not related to the COVID-19 [14] and also the limited supportive resources to handle the workload during the COVID-19 pandemic [2, 11].

## Conclusion

There is high prevalence of burnout among health workers caring for COVID-19 patients in Ghana. Night shifts and working at the primary level of care are associated with the highest rate of burnout. It is recommended that the Ministry of Health has to put in place policies to boost staff resilience and mitigate the experience of burnout among health workers now and beyond the pandemic such as scheduled psychological counseling for night shift staff and structured staff mentorship programmes for staff at primary care facilities. Also, longitudinal studies are needed to assess how health workers have over the period coped with the increased workload following the COVID-19 pandemic

## Supporting information

**S1 Table. Comparison of clinicians and non-clinicians on the socio-demographic and job-related factors.**
(DOCX)

**S1 File.**
(XLSX)

## Acknowledgments

Our profound appreciation to the staff of the three hospitals involved in the study. Special thanks to Miss Theresa Akua Appiah, Mr. Stephen Narkotey and Mr. Emmanuel Abindau for their support during the data collection.

## Author Contributions

**Conceptualization:** Kennedy Dodam Konlan.

**Data curation:** Emmanuel Asampong.

**Formal analysis:** Kennedy Dodam Konlan.

**Investigation:** Kennedy Dodam Konlan, Franklin N. Glozah.

**Methodology:** Emmanuel Asampong, Phyllis Dako-Gyeke.

**Supervision:** Emmanuel Asampong, Phyllis Dako-Gyeke, Franklin N. Glozah.

**Validation:** Franklin N. Glozah.

**Writing – original draft:** Kennedy Dodam Konlan.

**Writing – review & editing:** Emmanuel Asampong, Phyllis Dako-Gyeke.

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
