## [Decision Letter · Decision Letter 0]

23 Dec 2021

PONE-D-21-24220Burnout syndrome among health workers in human resource-constrained settings in urban Ghana: socio-demographic and job-related predictorsPLOS ONE

Dear Dr. Konlan,

Thank you for submitting your manuscript to PLOS ONE. After careful consideration, we feel that it has merit but does not fully meet PLOS ONE’s publication criteria as it currently stands. Therefore, we invite you to submit a revised version of the manuscript that addresses the points raised during the review process.

Specifically:

-Revising the title of the study to include COVID-19

-Stating the research questions clearly and in detail

-Paying more attention to the methods section (i.e., sample size calculation, inclusion/exclusion criteria, …)

-Paying more attention to the definitions of some variables and the process of categorizing them. -Also PLS describe how the categories of burnout were created.

-Merging tables and revising them according to the research questions.

-Discussing other potential risk factors of occupational burnout during the COVID-19 pandemic 

-Discussing potential sources of bias in the study

-Paying more attention to the usage of acronyms

We look forward to receiving your revised manuscript.

Kind regards,

Forough Mortazavi

Academic Editor

PLOS ONE

“This study was supported by the Carnegie Corporation of New York (CCNY), Building a New Generation of Academics in Africa (BANGA) Project Grant for Thesis Completion Award.”

Additional Editor Comments:

Dear authors,

Thank you for working on burnout during COVID-19 pandemic. This is an interesting and timely study. The authors collected data on several important variables related to burnout but the results of the study have not been presented in a satisfactory manner. In my opinion, the authors should define the research questions clearly at the beginning and focus on them in results section. For example, if the main aim of the study is to investigate predictors of burnout then there is no need to present the socio demographic data according to clinical/nonclinical state of the participants. Instead, they should consider clinical/nonclinical state of the participants as an independent variable in the tables. If the authors aimed to compare the level of burnout between clinician and non-clinician individuals, they should include p-value for comparisons in the tables.

As the authors mention in the manuscript, the fear of contracting COVID-19 in the workplace could lead to burnout. Therefore, I strongly recommend that the authors include 'COVID-19' in the title of their paper. The title proposed by one of the reviewers seems more interesting and appropriate with regard to the content of this study. I also recommend that the authors provide some information in the introduction section on the situation of the country during the study period in terms of COVID-19 infection/death rates.

Background and Conclusions in the abstract should be summarized.

The period of the study should be mentioned in the abstract.

The inclusion/exclusion criteria should be clearly stated.

Tables should be clearly laid out with no need for supplementary information. PLS spell out the abbreviations in the footnotes of tables.

Table 3 is suitable for a validity study of a scale.

The number of tables is unnecessarily high. Tables should be merged according to the research questions. If the authors’ aim is to present data for predictors of burnout then they should merge:

-tables 1 and 2

-tables 3, 4, 5, and 6

-tables 7 and 8

-tables 9 and 10.

I hope the authors take note of all my comments and improve the quality of the manuscript.

Reviewers' comments:

Reviewer's Responses to Questions

**Comments to the Author**

1. Is the manuscript technically sound, and do the data support the conclusions?

Reviewer #1: Yes

Reviewer #2: Yes

Reviewer #3: Yes

2. Has the statistical analysis been performed appropriately and rigorously? 

Reviewer #1: Yes

Reviewer #2: No

Reviewer #3: Yes

3. Have the authors made all data underlying the findings in their manuscript fully available?

Reviewer #1: Yes

Reviewer #2: No

Reviewer #3: Yes

4. Is the manuscript presented in an intelligible fashion and written in standard English?

Reviewer #1: Yes

Reviewer #2: No

Reviewer #3: Yes

5. Review Comments to the Author

Reviewer #1: I would like to thank you for inviting me to review this manuscript. In the present study, a cross-sectional study design was used to investigate the prevalence of burnout and its associated factors among health workers in human resource-constrained settings in urban Ghana. In my view, although the research idea is highly interesting, the study could be more precise. The following comments would be helpful to enhance the quality of the study to be endorsed by the scientific community.

1. Regardless of the results obtained from the model, it is crucial to emphasize that accurate predictions cannot be guaranteed by cross-sectional study. Rather, development of prediction models is based on cohort study. Thus, prediction models resulting from cross-sectional designs can be misleading. Therefore, it is necessary to consider this point in the interpretation of the results of this study. I recommend that the authors use the word association (or relationship) instead of prediction in the whole article.

2. The title of the article does not reflect the content appropriately. What do you think about the following title?

“Burnout among clinical and non-clinical health workers in Ghana during the COVID-19 pandemic: The roles of socio-demographic and job-related factors”

3. In the “limitations of the study” section, I suggest the authors to add some brief consideration on other potential risk factors of occupational burnout (e.g., grief caused by multiple losses, personal characteristics, and limited supportive resources to handle workload) during the COVID-19 pandemic with appropriate references (see doi: 10.1177/20101058211040575; doi: 10.3889/oamjms.2020.5502; doi: 10.1007/s12144-020-01292-0).

4. The overall conclusion does not rise to the level of conceptual advance. The “conclusion” section should be briefly described the implications of the study, followed by recommendations for future studies.

5. In page 25, please replace “Afulani et al. (2021a)” with “Afulani et al. [2].”

6. Please review the manuscript for use of the acronyms. In addition, define the acronyms separately in the abstract and introduction before using them in other sections of the manuscript. Moreover, the number of acronyms in the article is so large which makes it hard for the reader to follow. Please minimize the use of acronyms in all parts of the article, particularly in the abstract section.

7. Please add acronyms in the footnotes of tables.

Reviewer #2: Title of manuscript need revision in line with objective of study

in abstract, it should no more than 250 word

result need to revise

conclusion should be concise

in introduction

literature review should be recent and relevant with objective of study

rational of study should be strong

in method section

study setting should be explained more

sample size need explain how to calculate which software and parameter

study variable should be explain

exclusion criteria should be explain

study tool should be explain and reliability and validity should be include

statistical analysis should be revised and relevant statistical test should be apply

result of study should be revised according to revise test

in discussion need to revised with compare of relevant studies

limitation of study and bias of study should be add

references need to revise with latest after 2019

Reviewer #3: Please note the attachment. The manuscript should be revised as enclosed. It also requires further statistical analysis in order to validate this finding of this interesting study. Also References should follow Vancouver style as per the journal guidelines.

6. PLOS authors have the option to publish the peer review history of their article (what does this mean?). If published, this will include your full peer review and any attached files.

Reviewer #1: No

Reviewer #2: **Yes: **Mubashir Zafar

Reviewer #3: No

---

## [Author Response · Author response to Decision Letter 0]

16 Feb 2022

RESPONSE TO MANUSCRIPT COMMENTS

We are grateful that you considered our manuscript for review. We have provided a point by point response to each observation made by the reviewer.

Authors’ general comments

We are most grateful that you spent time within your important schedule to review and make insightful remarks regarding our manuscript. Your contributions and views are generally appreciated. We have incorporated the comments as you indicated within the manuscript.

Editor’s comments

The concerns of the editor are outlined below:

University of Ghana

College of Health Sciences

School of Public Health

Department of Social & Behavioural Sciences

4th February, 2022

-Revising the title of the study to include COVID-19 -Stating the research questions clearly and in detail

-Paying more attention to the methods section (i.e., sample size calculation, inclusion/exclusion criteria, ...)

-Paying more attention to the definitions of some variables and the process of categorizing them. -Also PLS describe how the categories of burnout were created.

-Merging tables and revising them according to the research questions.

-Discussing other potential risk factors of occupational burnout during the COVID-19 pandemic

-Discussing potential sources of bias in the study -Paying more attention to the usage of acronyms

Author’s responses to Editor’s comments

- The title of the manuscript has been revised to include COVID-19 as recommended by the editor (title page ie page 1)

- The research question has been stated clearly and in detail in page 4 .of the manuscript

- The authors have paid more attention to the methods section particularly stating the inclusion and erxclusion criteria in page 5 and stating the methods for deriving the sample size in page 5.

- The authors have clearly stated how the variables such as level of resilience, burnout among others were derived/defined on page 7 and 8 of the manuscript.

- The authors have merged the tables as recommended and reduced the number of tables from 11 tables to 6 tables.

- Other potential risk factors for burnout during the COVID-19 period have been discussed in the limitations of the study on page 23 with supporting references.

- The potential sources of bias have been discussed in the limitations of the study on page 23.

- The number of acronmys have been reduced and attention given to those in the manuscript as recommended.

Additional Editor Comments:

The concerns of the additional editor are outlined below:

 Thank you for working on burnout during COVID-19 pandemic. This is an interesting

 and timely study. The authors collected data on several important variables related

 to burnout but the results of the study have not been presented in a satisfactory

 manner. In my opinion, the authors should define the research questions clearly at

 the beginning and focus on them in results section. For example, if the main aim of

 the study is to investigate predictors of burnout then there is no need to present the

 socio demographic data according to clinical/nonclinical state of the participants.

 Instead, they should consider clinical/nonclinical state of the participants as an

 independent variable in the tables. If the authors aimed to compare the level of

 burnout between clinician and non-clinician individuals, they should include p-value

 for comparisons in the tables.

 As the authors mention in the manuscript, the fear of contracting COVID-19 in the

 workplace could lead to burnout. Therefore, I strongly recommend that the authors

 include 'COVID-19' in the title of their paper. The title proposed by one of the

 reviewers seems more interesting and appropriate with regard to the content of this

 study. I also recommend that the authors provide some information in the

 introduction section on the situation of the country during the study period in terms of

 COVID-19 infection/death rates.

 -Background and Conclusions in the abstract should be summarized

 -The period of the study should be mentioned in the abstract.

 -The inclusion/exclusion criteria should be clearly stated.

 -Tables should be clearly laid out with no need for supplementary information. PLS

 spell out the abbreviations in the footnotes of tables.

 -Table 3 is suitable for a validity study of a scale.

 -The number of tables is unnecessarily high. Tables should be merged according to

 the research questions. If the authors’ aim is to present data for predictors of burnout

 then they should merge:

 -tables 1 and 2

 -tables 3, 4, 5, and 6

 -tables 7 and 8

 -tables 9 and 10.

 I hope the authors take note of all my comments and improve the quality of the

 manuscript.

Authors response to additional editor’s comments

- The results have being re-organised and p-values for comparison of the socio- demographic and job-related factors have been present in Tables 1 and 2 on pages 10 and 12

- COVID-19 has been included in the title of the study as recommended

- Information on the COVID-19 situation in Ghana during the study period has been

provided in the last paragraph of the introduction on page 4 of the manuscript

- The background and conclusion in the abstract have been summarized as

recommended.

- The period of the study have been mentioned in the abstract (methods of abstract)

- The inclusion and exclusion criteria have been stated in the methods of the

manuscript on page on page 5 (selection of participants and data collection).

- Tables have been clearly laid out in the manuscript with appropriate footnotes

- Table 3 has been taken out as recommended

- The number of tables have been reduced and those mergers recommended have

been done.

Reviewer’s comments

 Reviewer #1: I would like to thank you for inviting me to review this manuscript. In

 the present study, a cross-sectional study design was used to investigate the

 prevalence of burnout and its associated factors among health workers in human

 resource-constrained settings in urban Ghana. In my view, although the research

 idea is highly interesting, the study could be more precise. The following comments

 would be helpful to enhance the quality of the study to be endorsed by the scientific

 community.

 1. Regardless of the results obtained from the model, it is crucial to emphasize that

 accurate predictions cannot be guaranteed by cross-sectional study. Rather,

 development of prediction models is based on cohort study. Thus, prediction models

 resulting from cross-sectional designs can be misleading. Therefore, it is necessary

 to consider this point in the interpretation of the results of this study. I recommend

 that the authors use the word association (or relationship) instead of prediction in the

 whole article.

 2. The title of the article does not reflect the content appropriately. What do you think

 about the following title?

 “Burnout among clinical and non-clinical health workers in Ghana during the COVID-

 19 pandemic: The roles of socio-demographic and job-related factors”

 3. In the “limitations of the study” section, I suggest the authors to add some brief

 consideration on other potential risk factors of occupational burnout (e.g., grief

 caused by multiple losses, personal characteristics, and limited supportive resources

 to handle workload) during the COVID-19 pandemic with appropriate references (see

 doi: 10.1177/20101058211040575; doi: 10.3889/oamjms.2020.5502; doi:

 10.1007/s12144-020-01292-0).

 4. The overall conclusion does not rise to the level of conceptual advance. The

 “conclusion” section should be briefly described the implications of the study,

 followed by recommendations for future studies.

 5. In page 25, please replace “Afulani et al. (2021a)” with “Afulani et al. [2].”

 6. Please review the manuscript for use of the acronyms. In addition, define the

 acronyms separately in the abstract and introduction before using them in other

 sections of the manuscript. Moreover, the number of acronyms in the article is so

 large which makes it hard for the reader to follow. Please minimize the use of

 acronyms in all parts of the article, particularly in the abstract section.

 7. Please add acronyms in the footnotes of tables.

Authors’ response to reviewer 1

-The authors have used association/relationship throughout the entire article as recommended - The title of the manuscirpt has been modified as recommended.

- The other risk factors of occupational burnout have been stated in the limitation section as recommended.

- The recommendeations on the conclusion section have been effected

- The suggestions on page 25 regarding the reference has been done

- the number of acronyms have been reduced and those used have been clearly defined

Reviwer no. 2 comments

 Reviewer #2: Title of manuscript need revision in line with objective of study

 in abstract, it should no more than 250 word

 result need to revise

 conclusion should be concise

 in introduction

 literature review should be recent and relevant with objective of study

 rational of study should be strong

 in method section

 study setting should be explained more

 sample size need explain how to calculate which software and parameter

 study variable should be explain

 exclusion criteria should be explain

 study tool should be explain and reliability and validity should be include

 statistical analysis should be revised and relevant statistical test should be apply

 result of study should be revised according to revise test

Authors’ responses to reviewer no.2’s comments

- The title of the manuscript has been revised in line with the objective of the study

- The results has been revised as recommended and number of tables reduced

- The conclusion has been made concise as recommended

- In the introduction, the literature used has been revised to more recent and rational

for the study stated as recommended

- In the method section, the method in which the sample size was derived has been

stated and study setting explained further. Also, the information on the study tool

 in discussion need to revised with compare of relevant studies

 limitation of study and bias of study should be add

 references need to revise with latest after 2019

and their relevant validity has been stated. Further, the appropriate statistical

approaches have been used to anaklyse the data.

- The results have been revised and this was used to revise the text.

- The disucssions have been compared with other relevant studies

- References have been revised and papers after 2019 inckuded as recommended.

Reviewer no.3 comments

PONE-D-21-24220

Burnout syndrome among health workers in human resource-constrained settings in urban Ghana: socio-demographic and job-related predictors

Comments:

Ethics statement: The authors stated that: the data in this paper is part of study titled “burnout, metabolic syndrome and predisposition to non-communicable diseases”. Also, this was mentioned in page 9 , second paragraph. Was the original study published? If yes, you need its reference?

Abstract: It is very long and should be rewritten. As per PLOS instructions to authors: authors should summarize the most important results and their significance and not exceed 300 words.

Methods: Explain in more details’ inclusion and exclusion criteria of the participants. What was the date and period of the study?

Page 6 paragraph 5: the authors stated that: “All participants who took part in this study were not directly involved in caring for patients with COVID-19..........” How about clinicians working in ER and ICU ....and other department caring for patients with COVID-19. Please clarify.

Explain” Veronica bucket” as a mechanism for hand washing originating in Ghana and readers are not familiar with it.

Under instrument: How long did it take to complete the questionnaires?

Level of Resilience: What were the scores for low, normal and high levels showed in Table 1?

Burnout syndrome: What are the scores in Masalah Burnout Inventory to indicate “no burnout” versus “burnout”? in other words to diagnose burnout syndrome? Furthermore, was the latter classified into mild, moderate, and severe? The authors stated that “Participants with high score in EE and depersonalization...” what are the levels of high, moderate, and low scores? Also, these were mentioned under Table 4.

How were the data extracted?

Results:

Authors should prove that all clinical and non-clinical participants were comparable in all possible confounding factors related to the outcome. Instead of reporting the total number column in Table 1, do statistical analysis to see if there were statistically significant between both groups.

What was the range of the participants’ age?

Under Prevalence of burnout among participants page 14: the authors stated “..... compared to the clinical participants, the non-clinical participants had a significantly higher prevalence of burnout” What was the p-value?

Tables: there are 10 Tables, large numbers. I suggest delete some of them such as Table 3 and instead mention the findings in the text. Table 4 and Table 5 can be merged under one Table and the same for

Table 7 with 9 and 8 with 10. Or just keep one of each similar Table and mention the positive finding of the other in the text.

Table 2: What were the definitions of night, afternoon, and morning work shifts? What were job support and job control and how did they measure them? Explain these in the text.

Discussion: on page 22, second paragraph authors sated that “The finding of high burnout rate could be because of the reduced resources resources......................... inadequate preparedness of health workers to handle emerging disease....” These were assumption and were not investigated in this study. I suggest deleting this paragraph.

Limitation of the study: explain in logical ways. This study was susceptible to survivor bias because it assesses prevalence rather than incident cases. It is also likely that all the scores were subjected to

reporting bias, since the data here were collected through a questionnaire.

References: PLOS uses “Vancouver” style. The manuscript presents several references which were outdated. The references should be revised, updated, and should follow the guidelines of the journal.

Authors’ responses to reviewer 3

- There is no publication yet from the study and this has been stated on page 9 on ethics In the manuscript.

- The length of the abstract has been reduced to less than 300 words as recommended

- The inclusion and exclusion criteria has been stated on page 5 of the manuscript.

- The date and period of the study has been stated in the abstract and on the last

paragraph of page 6 of the manuscript.

- The suggestions on page 6 of the manuscript have been effected.

- The clarifications on veronica bucket has been done.

- The duration for filling the instrument has been stated in the second sentence under

instrument on page 7.

- The scores under level of resilience have been stated on page 8 of the manuscript.

- The determination of burnout and the scores on the elements of the burnout

syndrome have been stated on page 8 under burnout syndrome.

- How the data was extracted is provided under data analysis on page 8 of the

manuscript.

- The comparison of the study participants have been made under table 1 as

suggested.

- The age range for the participants has been stated in page 10 of the manuscript.

- The p-value for the prevalence of burnout has been provided on page 13 of the

manuscript as recommended.

- The tables have been merged as recommended by the reviewer

- The definitions for night, afternoon and morning shifts have been defined on page

11 of the manuscript

- Also, the definitions for job control and job support have been provided on page 11

of the manuscript.

- The recommendations on the discussions, limitations and references have been

effected .

Conclusions

- We do hope that the corrections we have made in the manuscript meet the demands of the reviewer.

Yours sincerely, Kennedy Dodam Konlan

---

## [Decision Letter · Decision Letter 1]

21 Feb 2022

PONE-D-21-24220R1Burnout among clinical and non-clinical health workers in Ghana during the COVID-19

pandemic: The roles of socio-demographic and job-related factorsPLOS ONE

Dear Dr. Konlan,

Thank you for submitting your manuscript to PLOS ONE. After careful consideration, we feel that it has merit but does not fully meet PLOS ONE’s publication criteria as it currently stands. Therefore, we invite you to submit a revised version of the manuscript that addresses the points raised during the review process.

Specifically: 

-Revising the title of the study  

- The results in the abstract should be presented together with statistical measures such as OR, p-value or CI.

-Stating the research questions clearly in both the text and the abstract

-Statistical tests performed need to be stated in the methods section of the abstract. 

- Providing useful information about the settings of the study is important

- Merging tables 1-2 and revising them according to the research questions.  

- Bias and external validity of study should be described     

 Please submit your revised manuscript by Apr 07 2022 11:59PM. If you will need more time than this to complete your revisions, please reply to this message or contact the journal office at plosone@plos.org. Please include the following items when submitting your revised manuscript:A rebuttal letter that responds to each point raised by the academic editor and reviewer(s). You should upload this letter as a separate file labeled 'Response to Reviewers'.A marked-up copy of your manuscript that highlights changes made to the original version. You should upload this as a separate file labeled 'Revised Manuscript with Track Changes'.An unmarked version of your revised paper without tracked changes. You should upload this as a separate file labeled 'Manuscript'.

We look forward to receiving your revised manuscript.

Kind regards,

Forough Mortazavi

Academic Editor

PLOS ONE

Additional Editor Comments:

Dear authors,

Thank you for revising the manuscript. Several comments have been addressed satisfactorily but there are still some comments which you have not responded to. The results of the study still have not been presented in a satisfactory manner. In my opinion, the authors should define the research questions clearly at the beginning and then deal with them in results and discussion sections. For example, if the main aim of the study is to investigate predictors of burnout, as the authors seem to imply in tables 4-6, the key words, the discussion section, and the abstract, then there is no need to present the socio demographic data according to clinical/nonclinical state of the participants. Instead, they should regard clinical/nonclinical state of the participants as an independent variable in the tables. It is quite clear from the contents of the tables 4-6 what the main aim of the study is and so tables 1-2 must be consistent with the main aim of the study as presented by tables 4-6. In my opinion, the comparisons of clinicians and none clinicians (tables 1-2) should only be presented as supplementary information. Socio-demographic and job-profiles of participants should be presented in a single table covering all the sample members.

The title proposed by one of the reviewers seems more informative and appropriate with regard to the contents of the study: “burnout syndrome among healthcare workers during COVID-19 pandemic”

The remarks on “Aims” in both the abstract and the text are in need of revision.

Limitations should be included in the abstract.

I hope the authors take note of all my comments and the comments raised by reviewers and improve the quality of the manuscript.

Reviewers' comments:

Reviewer's Responses to Questions

**Comments to the Author**

1. If the authors have adequately addressed your comments raised in a previous round of review and you feel that this manuscript is now acceptable for publication, you may indicate that here to bypass the “Comments to the Author” section, enter your conflict of interest statement in the “Confidential to Editor” section, and submit your "Accept" recommendation.

Reviewer #1: All comments have been addressed

Reviewer #2: (No Response)

Reviewer #3: All comments have been addressed

2. Is the manuscript technically sound, and do the data support the conclusions?

Reviewer #1: Yes

Reviewer #2: Partly

Reviewer #3: Yes

3. Has the statistical analysis been performed appropriately and rigorously? 

Reviewer #1: Yes

Reviewer #2: No

Reviewer #3: Yes

4. Have the authors made all data underlying the findings in their manuscript fully available?

Reviewer #1: Yes

Reviewer #2: No

Reviewer #3: Yes

5. Is the manuscript presented in an intelligible fashion and written in standard English?

Reviewer #1: Yes

Reviewer #2: Yes

Reviewer #3: Yes

6. Review Comments to the Author

Reviewer #1: Dear Authors,

I carefully read the revised manuscript.

I would like to thank you very much indeed for the modifications you have made to the newly submitted manuscript

Kind regards

Reviewer #2: Title of study need to revise as burnout syndrome among Healthcare workers during COVID-19 pandemic

in abstract

method section need to revise as which statistical test apply, in result section finding should be mentioned in numbers

Rational of study should be strong in the introduction section

study setting need to explain more as how many patients coming to selected hospitals daily and how many doctors available

sample size need to re-calculate

chi square test not appropriate test to determine the association

it need to apply regression test

bias of study should be explain

external validity of study should be explain

Reviewer #3: The required corrections have been made and the manuscript has been improved scientifically. The authors responded to my comment and provided valuable information which are important for the readers.

7. PLOS authors have the option to publish the peer review history of their article (what does this mean?). If published, this will include your full peer review and any attached files.

Reviewer #1: No

Reviewer #2: No

Reviewer #3: No

---

## [Author Response · Author response to Decision Letter 1]

23 Mar 2022

University of Ghana

College of Health Sciences

School of Public Health

Department of Social & Behavioural

Sciences

23rd March, 2022

The Editor

PLos One

Dear Sir/Madam,

RESPONSE TO MANUSCRIPT COMMENTS

We are grateful that you considered our manuscript for review. We have provided a point by point response to each observation made by the reviewer.

Authors’ general comments

We are most grateful that you spent time within your important schedule to review and make insightful remarks regarding our manuscript. Your contributions and views are generally appreciated. We have incorporated the comments as you indicated within the manuscript.

Editor’s comments

The concerns of the editor are outlined below:

-Revising the title of the study 

- The results in the abstract should be presented together with statistical measures such as OR, p-value or CI. 

-Stating the research questions clearly in both the text and the abstract 

-Statistical tests performed need to be stated in the methods section of the abstract. 

- Providing useful information about the settings of the study is important 

- Merging tables 1-2 and revising them according to the research questions. 

- Bias and external validity of study should be described

Author’s responses to Editor’s comments

- The title of the manuscript has been revised as recommended by the editor (title page ie page 1)

- The results in the abstract have been modified to include statistical measures such as OR, p-value or CI as recommended (page 2).

- The research question has been clearly stated in both the abstract and text as recommended. 

- The statistical tests that were performed have been stated in the methods section of the abstract (page 2).

- The authors have provided additional information on the setting of the study has been provided on page 5 of the manuscript as suggested. 

- Tables 1 and 2 have been merged into a single table as recommended.

- The bias and external validity of the study have been explained on page 23 as recommended. 

Additional Editor Comments:

The concerns of the additional editor are outlined below:

-Socio-demographic and job-profiles of participants should be presented in a single table covering all the sample members. 

-The title proposed by one of the reviewers seems more informative and appropriate with regard to the contents of the study: “burnout syndrome among healthcare workers during COVID-19 pandemic” 

-The remarks on “Aims” in both the abstract and the text are in need of revision. 

-Limitations should be included in the abstract. 

I hope the authors take note of all my comments and the comments raised by reviewers and improve the quality of the manuscript.

Authors’ response to additional editor’s comments

- The results have being re-organised and Tables 1 and 2 have ben merged as suggested.

- The title of the study has been revised as suggested.

- The aim has been revised as recommended

- The limitation has been written properly and included in the abstract. 

Reviewer’s comments

Reviewer #2: 

-Title of study need to revise as burnout syndrome among Healthcare workers during COVID-19 pandemic 

-in abstract method section need to revise as which statistical test apply, 

-in result section finding should be mentioned in numbers 

-Rational of study should be strong in the introduction section

- study setting need to explain more as how many patients coming to selected hospitals daily and how many doctors available 

-sample size need to re-calculate

- chi square test not appropriate test to determine the association it need to apply regression test 

-bias of study should be explain external validity of study should be explain.

Authors’ response to reviewer 2

-The title of the study has been revised as suggested.

- The statistical analysis that was done has been stated in the methods section of the abstract.

- The results section has been modified and specific numbers have been mentioned as recommended. 

- The rationale for the study has been well stated in the introduction.

-The study settings has been explained further as recommended 

- The rationale for the sample size has been explained. 

- Regression analysis has been added in addition to the chi-square test.

-Bias and external validity of the study has been explained in the limitations of the study.

Conclusions

- We do hope that the corrections we have made in the manuscript meet the demands of the reviewer.

Yours sincerely,

Kennedy Dodam Konlan

---

## [Decision Letter · Decision Letter 2]

28 Mar 2022

PONE-D-21-24220R2Burnout Syndrome Among Healthcare workers During COVID-19 PandemicPLOS ONE

Dear Dr. Konlan,

Thank you for submitting your manuscript to PLOS ONE. After careful consideration, we feel that it has merit but does not fully meet PLOS ONE’s publication criteria as it currently stands. Therefore, we invite you to submit a revised version of the manuscript that addresses the points raised during the review process.

Specifically: 

- Appropriate statistical tests as noted by the reviewer should be performed.

- The percentage of burnout and how it is calculated should be clarified and presented in detail.

We look forward to receiving your revised manuscript.

Kind regards,

Forough Mortazavi

Academic Editor

PLOS ONE

Additional Editor Comments:

Dear authors,

Thank you for revising the manuscript. Several comments have been addressed satisfactorily but the manuscript is still in need of further revision. In my opinion, the results of the study have not been presented in the manuscript in a precise manner. I hope the authors take note of all my comments and the comments raised by the reviewer and improve the quality of the manuscript.

- In page 9 the authors state, “Participants with high scores on emotional exhaustion and depersonalization as well as low scores on personal accomplishment sub-scales were diagnosed as having burnout [24, 26].” But the authors’ criteria for designating a participant as having experienced burnout has not been clearly argued and justified and it seems arbitrary and inconsistent. The authors should determine the percentage of participant who experienced burnout.

- No information is presented regarding the number of participants with high scores on emotional exhaustion and depersonalization and low scores on personal accomplishment sub-scales. I think the authors should start with calculating the total score for this scale and then use it in linear regression analysis.

- PLS do not repeat the results both in text and in tables.

Reviewers' comments:

Reviewer's Responses to Questions

**Comments to the Author**

1. If the authors have adequately addressed your comments raised in a previous round of review and you feel that this manuscript is now acceptable for publication, you may indicate that here to bypass the “Comments to the Author” section, enter your conflict of interest statement in the “Confidential to Editor” section, and submit your "Accept" recommendation.

Reviewer #1: All comments have been addressed

Reviewer #2: All comments have been addressed

Reviewer #3: All comments have been addressed

2. Is the manuscript technically sound, and do the data support the conclusions?

Reviewer #1: Yes

Reviewer #2: No

Reviewer #3: Yes

3. Has the statistical analysis been performed appropriately and rigorously? 

Reviewer #1: Yes

Reviewer #2: No

Reviewer #3: Yes

4. Have the authors made all data underlying the findings in their manuscript fully available?

Reviewer #1: Yes

Reviewer #2: No

Reviewer #3: Yes

5. Is the manuscript presented in an intelligible fashion and written in standard English?

Reviewer #1: Yes

Reviewer #2: No

Reviewer #3: Yes

6. Review Comments to the Author

Reviewer #1: Dear authors,

You have done an impressive job.

I carefully checked the manuscript for any problematic parts. Luckily I could not find any.

The manuscript is well-written. The data analyses have been conducted accurately. Moreover, plausible inferences have been made according to the results and findings of the study.

Kind regards,

Reviewer #2: It need to revise analysis

Linear equations regression should apply . limitations of study should be add

Discussion need revised

Reviewer #3: The required corrections have been made and the manuscript has been improved scientifically. The authors responded to my comment and provided valuable information which are important for the readers.

7. PLOS authors have the option to publish the peer review history of their article (what does this mean?). If published, this will include your full peer review and any attached files.

Reviewer #1: No

Reviewer #2: No

Reviewer #3: No

---

## [Author Response · Author response to Decision Letter 2]

29 Mar 2022

University of Ghana

College of Health Sciences

School of Public Health

Department of Social & Behavioural

Sciences

29th March, 2022

The Editor

PLos One

Dear Sir/Madam,

RESPONSE TO MANUSCRIPT COMMENTS

We are grateful that you considered our manuscript for review. We have provided a point by point response to each observation made by the reviewer.

Authors’ general comments

We are most grateful that you spent time within your important schedule to review and make insightful remarks regarding our manuscript. Your contributions and views are generally appreciated. We have incorporated the comments as you indicated within the manuscript.

Editor’s comments

The concerns of the editor are outlined below:

Specifically: 

- Appropriate statistical tests as noted by the reviewer should be performed. 

- The percentage of burnout and how it is calculated should be clarified and presented in detail.

Author’s responses to Editor’s comments

- We have used the appropriate statistical test appropriate for the data that was collected. The outcome variable in this study was burnout which was classified as a categorical variable and hence logistic regression been appropriate for the analysis. Linear equations regressions as suggested by the reviewer is not appropriate when dealing with a categorical outcome variable. 

- The percentage of burnout and how it was calculated has been clarified and presented in page 13 of the revised manuscript.

Additional Editor Comments:

The concerns of the additional editor are outlined below:

- In page 9 the authors state, “Participants with high scores on emotional exhaustion and depersonalization as well as low scores on personal accomplishment sub-scales were diagnosed as having burnout [24, 26].” But the authors’ criteria for designating a participant as having experienced burnout has not been clearly argued and justified and it seems arbitrary and inconsistent. The authors should determine the percentage of participant who experienced burnout.

 - No information is presented regarding the number of participants with high scores on emotional exhaustion and depersonalization and low scores on personal accomplishment sub-scales. I think the authors should start with calculating the total score for this scale and then use it in linear regression analysis. 

- PLS do not repeat the results both in text and in tables

Authors’ response to additional editor’s comments

- The authors have provided the diagnostic criterion for burnout on page 9 of the revised manuscript and supported same by literature. And the percentage of those with burnout stated on page 13 of the revised manuscript as requested. 

- Information on participants with high scores on emotional exhaustion, depersonalization and personal accomplishment have been presented on page 13 of the revised manuscript (Table 2).

- Respectfully, the data for the study particularly the outcome variable (burnout) was categorical and thus not practicable to use linear regression analysis as argued. Linear regression analysis is used when the output or the dependent variable is continuous. Such models are used for risk assessment and predictions and this was not the focus of this study. We thus used the logistic regression analysis for our categorical outcome variable. 

- It is thus properly so that we used logistic regression models in our analysis since that is appropriate for categorical outcome variables. 

- We have ensured no repeat of results in text and tables. 

Reviewer’s comments

Reviewer #2:

- It need to revise analysis 

-Linear equations regression should apply. 

-limitations of study should be add 

-Discussion need revised

Authors’ response to reviewer 2

-We have revised the analysis as suggested 

-Burnout in this study was classified as a categorical variable hence linear equation regression analysis cannot be applied as suggested. 

-Limitations of the study have been added in the discussion section on page 24 of the revised manuscript.

-Discussions have been revised appropriately 

Conclusions

- We do hope that the corrections we have made in the manuscript meet the demands of the editor and reviewer.

Yours sincerely,

Kennedy Dodam Konlan

---

## [Decision Letter · Decision Letter 3]

8 Apr 2022

PONE-D-21-24220R3Burnout Syndrome Among Healthcare workers During COVID-19 PandemicPLOS ONE

Dear Dr. Konlan,

Thank you for submitting your manuscript to PLOS ONE. After careful consideration, we feel that it has merit but does not fully meet PLOS ONE’s publication criteria as it currently stands. Therefore, we invite you to submit a revised version of the manuscript that addresses the points raised during the review process.

We look forward to receiving your revised manuscript.

Kind regards,

Forough Mortazavi

Academic Editor

PLOS ONE

Journal Requirements:

Reviewers' comments:

Reviewer's Responses to Questions

**Comments to the Author**

1. If the authors have adequately addressed your comments raised in a previous round of review and you feel that this manuscript is now acceptable for publication, you may indicate that here to bypass the “Comments to the Author” section, enter your conflict of interest statement in the “Confidential to Editor” section, and submit your "Accept" recommendation.

Reviewer #2: All comments have been addressed

2. Is the manuscript technically sound, and do the data support the conclusions?

Reviewer #2: Partly

3. Has the statistical analysis been performed appropriately and rigorously? 

Reviewer #2: No

4. Have the authors made all data underlying the findings in their manuscript fully available?

Reviewer #2: Yes

5. Is the manuscript presented in an intelligible fashion and written in standard English?

Reviewer #2: No

6. Review Comments to the Author

Reviewer #2: Need to update reference

Update reference from 2019

Limitations of study should be add

Rational of study need revised

7. PLOS authors have the option to publish the peer review history of their article (what does this mean?). If published, this will include your full peer review and any attached files.

Reviewer #2: **Yes: **Mubashir Zafar

---

## [Author Response · Author response to Decision Letter 3]

9 Apr 2022

University of Ghana

College of Health Sciences

School of Public Health

Department of Social & Behavioural

Sciences

9th April, 2022

The Editor

PLos One

Dear Sir/Madam,

RESPONSE TO MANUSCRIPT COMMENTS

We are grateful that you considered our manuscript for review. We have provided a point by point response to each observation made by the reviewer.

Authors’ general comments

We are most grateful that you spent time within your important schedule to review and make insightful remarks regarding our manuscript. Your contributions and views are generally appreciated. We have incorporated the comments as you indicated within the manuscript.

Editor’s comments

The concerns of the editor are outlined below:

Author’s responses to Editor’s comments

- We have reviewed the reference list to ensure they are complete and correct as directed by the editors.

.

Reviewer’s comments

Reviewer #2:

- Need to update reference. Update reference from 2019.

- Limitations of study should be add

- Rational of study need revised

Authors’ response to reviewer 2

-We have updated the references on pages 25 to 28 of the revised manuscript and included appropriate and current references as suggested by the reviewer.

-We have added to the limitations of the study on page 24 of the revised manuscript 

-We have revised the rationale of the study on page 4 of the revised manuscript. 

Conclusions

- We do hope that the corrections we have made in the manuscript meet the demands of the editor and reviewer.

Yours sincerely,

Kennedy Dodam Konlan

---

## [Decision Letter · Decision Letter 4]

18 Apr 2022

PONE-D-21-24220R4Burnout Syndrome Among Healthcare workers During COVID-19 PandemicPLOS ONE

Dear Dr. Konlan,

Thank you for submitting your manuscript to PLOS ONE. After careful consideration, we feel that it has merit but does not fully meet PLOS ONE’s publication criteria as it currently stands. Therefore, we invite you to submit a revised version of the manuscript that addresses the points raised during the review process.

We look forward to receiving your revised manuscript.

Kind regards,

Forough Mortazavi

Academic Editor

PLOS ONE

Reviewers' comments:

Reviewer's Responses to Questions

**Comments to the Author**

1. If the authors have adequately addressed your comments raised in a previous round of review and you feel that this manuscript is now acceptable for publication, you may indicate that here to bypass the “Comments to the Author” section, enter your conflict of interest statement in the “Confidential to Editor” section, and submit your "Accept" recommendation.

Reviewer #2: All comments have been addressed

2. Is the manuscript technically sound, and do the data support the conclusions?

Reviewer #2: Partly

3. Has the statistical analysis been performed appropriately and rigorously? 

Reviewer #2: No

4. Have the authors made all data underlying the findings in their manuscript fully available?

Reviewer #2: No

5. Is the manuscript presented in an intelligible fashion and written in standard English?

Reviewer #2: Yes

6. Review Comments to the Author

Reviewer #2: Its need major revision

Tittle of study should be revised, should mentioned the city and country

Results in the abstract should be concise, only important result mentioned in abstract

Rationale of study still not good, explain what benefit to society, health care workers and non clinicians, policy makers

in study setting, number of patients admitted in hospitals and OPD should explain

sample size should be calculated through EPI info software

sampling technique should be revised

Please send the questionnaire for this study for evaluation of results

7. PLOS authors have the option to publish the peer review history of their article (what does this mean?). If published, this will include your full peer review and any attached files.

Reviewer #2: **Yes: **Mubashir Zafar

---

## [Author Response · Author response to Decision Letter 4]

20 Apr 2022

University of Ghana

College of Health Sciences

School of Public Health

Department of Social & Behavioural

Sciences

19th April, 2022

The Editor

PLos One

Dear Sir/Madam,

RESPONSE TO MANUSCRIPT COMMENTS

We are grateful that you considered our manuscript for review. We have provided a point by point response to each observation made by the reviewer.

Authors’ general comments

We are most grateful that you spent time within your important schedule to review and make insightful remarks regarding our manuscript. Your contributions and views are generally appreciated. We have incorporated the comments as you indicated within the manuscript.

Reviewer’s comments

Reviewer #2: Its need major revision 

Tittle of study should be revised, should mentioned the city and country 

Results in the abstract should be concise, only important result mentioned in abstract Rationale of study still not good, explain what benefit to society, health care workers and non clinicians, policy makers 

in study setting, number of patients admitted in hospitals and OPD should explain 

sample size should be calculated through EPI info software 

sampling technique should be revised 

Please send the questionnaire for this study for evaluation of results

Authors’ response to reviewer 2

-We have revised the title of the study to include the city and country as suggested by the reviewer (page 1 of the revised manuscript).

-The results in the abstract has been revised to include only relevant results (page 2 of revised manuscript).

- The rationale for the study has been revised as suggested by the reviewer (page 4 of revised manuscript).

-In the study setting, we have included the number of patients admitted in each hospital and at the OPD (pages 5 and 6 of the revised manuscript).

- We have stated the rationale for our choice of the sample size determination method we used on page 6 of the revised manuscript. Our explanation give credence to why EPI info software was not used. 

- We have revised the sampling technique on page 7 of the revised manuscript. 

- We have included the questionnaire that guided the study as a supporting documents. As stated in the Ethical consideration of the revised manuscript, the study was part of a bigger study titled 

“Burnout, metabolic syndrome and predisposition to non-communicable diseases: A study among health workers in the Greater Accra Region of Ghana”. Part of the data in this study is what this paper is communicating. 

Conclusions

- We do hope that the corrections we have made in the manuscript meet the demands of the editor and reviewer.

Yours sincerely,

Kennedy Dodam Konlan

---

## [Decision Letter · Decision Letter 5]

29 Apr 2022

Burnout Syndrome Among Healthcare workers During COVID-19 Pandemic in Accra, Ghana

PONE-D-21-24220R5

Dear Dr. Konlan,

We’re pleased to inform you that your manuscript has been judged scientifically suitable for publication and will be formally accepted for publication once it meets all outstanding technical requirements.

Kind regards,

Forough Mortazavi

Academic Editor

PLOS ONE

Additional Editor Comments (optional):

Reviewers' comments:

Reviewer's Responses to Questions

**Comments to the Author**

1. If the authors have adequately addressed your comments raised in a previous round of review and you feel that this manuscript is now acceptable for publication, you may indicate that here to bypass the “Comments to the Author” section, enter your conflict of interest statement in the “Confidential to Editor” section, and submit your "Accept" recommendation.

Reviewer #2: All comments have been addressed

2. Is the manuscript technically sound, and do the data support the conclusions?

Reviewer #2: No

3. Has the statistical analysis been performed appropriately and rigorously? 

Reviewer #2: No

4. Have the authors made all data underlying the findings in their manuscript fully available?

Reviewer #2: No

5. Is the manuscript presented in an intelligible fashion and written in standard English?

Reviewer #2: No

6. Review Comments to the Author

Reviewer #2: In abstract

Methods section need concise

Only major point included in the section

results need revised

Questionnaire need to send me because I will validate your results

Analysis needed to be revised

Appropriate test need to apply

Biased of study should be explain

7. PLOS authors have the option to publish the peer review history of their article (what does this mean?). If published, this will include your full peer review and any attached files.

Reviewer #2: **Yes: **Mubashir Zafar

---

## [Editor Report · Acceptance letter]

18 May 2022

PONE-D-21-24220R5 

Burnout Syndrome Among Healthcare workers During COVID-19 Pandemic in Accra, Ghana 

Dear Dr. Konlan:

I'm pleased to inform you that your manuscript has been deemed suitable for publication in PLOS ONE. Congratulations! Your manuscript is now with our production department. 

Kind regards, 

on behalf of

Dr. Forough Mortazavi 

Academic Editor

PLOS ONE